# INTEGRATION FLOW MODELS

## ABSTRACT

Recently, ordinary differential equation (ODE) based generative models have emerged as a cutting-edge method for producing high-quality samples in many applications. Generally, these methods typically involve learning continuous transformation trajectories that map a simple initial distribution (i.e., Gaussian noise) to the target data distribution (i.e., images) by multiple steps of solving different ODE functions in inference to obtain high-quality results. However, the ODE-based methods either suffer the discretization error of numerical solvers of ODE, which restricts the quality of samples when only a few NFEs are used, or struggle with training instability. In this paper, we proposed Integration Flow, which learns the results of ODE-based trajectory paths directly without solving the ODE functions. Moreover, Integration Flow explicitly incorporates the target state $\mathbf{x}_0$ as the anchor state in guiding the reverse-time dynamics and we have theoretically proven this can contribute to both stability and accuracy. To the best of our knowledge, Integration Flow is the first model with the unified structure to estimate ODE-based generative models. Through theoretical analysis and empirical evaluations, we show that Integration Flows achieve improved performance when it is applied to existing ODE-based model, such as diffusion models, Rectified Flows, and PFGM++. Specifically, Integration Flow achieves one-step generation on CIFAR10 with FID of 2.63 for Variance Exploding (VE) diffusion model, 3.4 for Rectified Flow without relflow and 2.96 for PFGM++. By extending the sampling to 1000 steps, we further reduce FID score to 1.71 for VE, setting state-of-the-art performance.

## 1 INTRODUCTION

Recently, ordinary differential equation (ODE) based generative models have emerged as a cutting-edge method for producing high-quality samples in many applications including image, audio (Kong et al., 2021; Popov et al., 2022), and video generation (Rombach et al., 2022; Saharia et al., 2022; Ho & Salimans, 2022). Generally, these methods typically involve learning continuous transformation trajectories that map a simple initial distribution (i.e. Gaussian noise) to the target data distribution (i.e. images) by solving ODEs (Figure 1).

Among those ODE-based models, the diffusion models have attracted the most attention due to their exceptional ability to generate realistic samples. The diffusion models employ a forward process that gradually adds noise to the data and a reverse process that reconstructs the original data by gradually removing the noise. To enhance sampling effectiveness, this reverse process is often reformulated as a probability flow Ordinary Differential Equations (PF-ODEs)(Song et al., 2020b). Despite their success, PF-ODE-based diffusion models face drawbacks due to their iterative nature, leading to high computational costs and prolonged sampling times during inference.

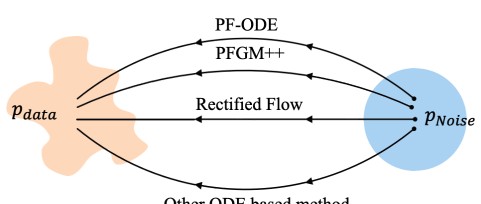

Figure 1: An illustration of ODE-based methods, including PF-ODE, PFGM++, and Rectified Flow.

Another ODE-based approach, rectified flow models (Liu et al., 2022; Lipman et al., 2022), aims to model the transformation between distributions via neural ODEs. These models focus on learning

smoother ODE trajectories that are less prone to truncation errors during numerical integration. By reducing the curvature of the generative paths, rectified flow enhances sampling efficiency and decreases the computational burden. However, even with smoother trajectories, rectified flow models still require considerable iterations to produce high-quality samples.

Building on the flow-based paradigm, Poisson Flow Generative Models (PFGM) and their extension PFGM++ have been introduced (Xu et al., 2022; 2023). Inspired by concepts from electrostatics, PFGM++ embeds data into a higher-dimensional space, specifically, an $N + D$ dimensional space where $D$ is the number of augmented dimensions. The generative process involves solving an ODE derived from the Poisson equation, tracing a path from a simple initial distribution (e.g., noise on a large hemisphere) to the target data distribution residing on a lower dimensional hyperplane. Similar to diffusion models and Rectified Flow, PFGM++ requires multiple steps during inference,

All aforementioned methods required multiple steps of solving different ODE functions in inference to obtain high-quality results. Furthermore, the ODE-based models naturally inherit the discretization error of numerical solvers of ODE, which restricts the quality of samples when only a few NFEs are used, or struggle with training instability when neural ODEs are used to approximate the ODE solution using neural networks. Given these challenges associated with ODE-based models, a natural question arises: can we learn the result of ODE-based trajectory paths directly without solving the ODE functions? Therefore, we can take the ODE function-defined generative model and solve this without an ODE solver. The answer is yes. Here we proposed **Integration Flow**, to the best of our knowledge, the first model with the unified structure to estimate ODE-based generative models.

Integration Flows represent a new type of generative models. Unlike traditional ODE-based approaches that focus on approximating the instantaneous drift term of an ODE or depend on iterative sampling methods, Integration Flows directly estimate the integrated effect of the cumulative transformation dynamics over time. This holistic approach allows for the modeling of the entire generative path in a single step, bypassing the accumulation of errors associated with high-curvature trajectories and multiple function evaluations. Integration Flows do not employ the ODE solver and eliminate the need for multiple sampling iterations, significantly reducing computational costs and enhancing efficiency.

Moreover, to increase the training stability and accuracy in reconstructing, Integration Flow explicitly incorporating the target state $\mathbf{x}_0$ as the anchor state in guiding the reverse-time dynamics from the intermediate state $\mathbf{x}_t$. We have theoretically proven that incorporating the target state $\mathbf{x}_0$ as the anchor state can provide a better or at least equal accurate estimation of $\mathbf{x}_0$.

In summary, Integration Flows addresses the limitations of existing ODE-based generative models by providing a unified and efficient approach to model the transformation between distributions. Our contributions can be outlined as follows:

- Introduction of Integration Flows: We present Integration Flows, a novel generative modeling framework that estimates the integrated dynamics of continuous-time processes without relying on iterative sampling procedures or traditional ODE solvers, that is, it supports one-step generation for (any) ODE-based generative models.

- Unified ODE-Based Generative Modeling: Integration Flows can adapt different ODE-based generative process, offering flexibility and unification across different generative modeling approaches.

- Enhanced Sampling Efficiency and Scalability: Through empirical evaluations, we show that Integration Flows achieve improved sampling efficiency and scalability compared to existing ODE-based models, such as diffusion models, Rectified Flows, and PFGM++. Specially, we set the state-of-the-art performance for one-step generation using Rectified Flow and PFGM++.

## 2 BACKGROUND AND RELATED WORKS

**Variance Exploding (VE) Diffusion Model**. The forward process in the Variance Exploding (VE) diffusion model(Song et al., 2020b; Karras et al., 2022) adds noise to the data progressively. This process is described as:

$$\mathbf{x}_t = \mathbf{x}_0 + \sigma_t \boldsymbol{\epsilon}, \quad t \in [0, T]$$

where $\mathbf{x}_0 \sim p_{\text{data}}$, $\sigma_t$ denotes noise schedule that increases with time $t$, $\boldsymbol{\epsilon} \sim \mathcal{N}(0, \mathbf{I})$.

The reverse process aims to denoise the data by starting from a noisy sample $\mathbf{x}_T$ and evolving it back to the clean data distribution $p_{\text{data}}$. This is achieved using the PF-ODEs, which models the continuous denoising process in the reverse direction. The PF-ODE is given by:

$$\frac{d\mathbf{x}_t}{dt} = -\frac{1}{2}\frac{d\sigma_t^2}{dt}\nabla_{\mathbf{x}_t}\log p_t\left(\mathbf{x}_t\right)$$

where $\nabla_{\mathbf{x}_t}\log p_t\left(\mathbf{x}_t\right)$ is the score function, representing the gradient of the log-probability of the data distribution $p_t\left(\mathbf{x}_t\right)$ at time $t$. $\frac{d\sigma_t^2}{dt}$ is the time derivative of the noise variance function $\sigma_t^2$, which controls how fast the noise is reduced as we reverse the process.

**Rectified Flows**. (Liu et al., 2022; Albergo & Vanden-Eijnden, 2022; Lipman et al., 2022) uses linear interpolation to connect the data distribution $p_{data}$ and a standard normal distribution $p_{\mathbf{z}}$ by introducing a continuous forward process that smoothly transitions between these two distributions, which is defined as:

$$\mathbf{z}_t = (1 - t)\mathbf{x}_0 + t\mathbf{z}, \quad t \in [0, 1]$$

where $\mathbf{x}_0$ is a sample drawn from the data distribution $p_{data}$, $\mathbf{z}$ is sampled from the standard normal distribution. This interpolation ensures that at $t = 0$, it recovers the original data point, i.e., $\mathbf{z}_0 = \mathbf{x}_0$, and at $t = 1$, the point has been mapped entirely into the noise distribution, i.e., $\mathbf{z}_1 = \mathbf{z}$. Thus, a straight path is created between the data and the noise distributions.

Liu et al.(Liu et al., 2022) demonstrated that for $\mathbf{z}_0 \sim p_{\mathbf{x}}$, the dynamics of the following ODE produce marginals that match the distribution of $\mathbf{x}_t$ for any $t$ :

$$\frac{d\mathbf{z}_t}{dt} = \mathbf{v}\left(\mathbf{z}_t, t\right)$$

Since the interpolation ensures that $\mathbf{x}_1 = \mathbf{z}$, the forward ODE transports samples from the data distribution $p_{\mathbf{x}}$ to the noise distribution $p_{\mathbf{z}}$. To reverse this process, starting with $\mathbf{z}_1 \sim p_{\mathbf{z}}$, the ODE can be integrated backward from $t = 1$ to $t = 0$, ultimately reconstructing samples from the data distribution.

**Poisson Flow Generative Models (PFGM)** PFGM++ (Xu et al., 2023) is a generalization of PFGM(Xu et al., 2022) that embeds generative paths in a high-dimensional space. It reduces to PFGM when $D = 1$ and to diffusion models when $D \to \infty$.

In PFGM++, each data point $\mathbf{x} \in \mathbb{R}^N$ is augmented by additional variables $\mathbf{z} = (z_1, \ldots, z_D) \in \mathbb{R}^D$, resulting in an augmented data representation $\tilde{\mathbf{x}} = (\mathbf{x}, \mathbf{z}) \in \mathbb{R}^{N+D}$. Due to the rotational symmetry of the electric field in the augmented space, the problem can be simplified by considering only the radial norm $r = \|\mathbf{z}\|_2$. This reduces the augmented data representation to $\tilde{\mathbf{x}} = (\mathbf{x}, r)$, where $r$ acts as a scalar anchor variable.

The electric field $\mathbf{E}(\tilde{\mathbf{x}})$ drives the dynamics of the generative process and can be decomposed into two key components:

$$\mathbf{E}(\tilde{\mathbf{x}}) = \left(\mathbf{E}(\tilde{\mathbf{x}})_{\mathbf{x}}, \mathbf{E}(\tilde{\mathbf{x}})_r\right),$$

where $\mathbf{E}(\tilde{\mathbf{x}})_{\mathbf{x}}$ represents the component of the electric field in the data space (i.e., along the original data dimensions $\mathbf{x}$), and $E(\tilde{\mathbf{x}})_r$ denotes the radial component in the augmented space. They are used to formulate the generative ODE.

Using the radial symmetry of the electric field, the backward ODE that governs the generative process can be expressed as:

$$\frac{d\mathbf{x}}{dr} = \frac{\mathbf{E}(\tilde{\mathbf{x}})_{\mathbf{x}}}{E(\tilde{\mathbf{x}})_r}$$

By solving this ODE in reverse, one can transport points from the high dimensional augmented space back to the original data space, completing the generative process.

## 3 METHOD

In this section, we will introduce the Integration Flow Models based on the general form of ODE-based generative models.

### 3.1 GENERAL FORM OF ODE-BASED GENERATIVE MODELS

Consider an initial state $\mathbf{x}_T$ drawn from a distribution $p(\mathbf{x}_T)$, typically chosen to be a simple distribution such as a Gaussian. The goal is to estimate $\mathbf{x}_0$, which is aligned with the data distribution $p_{\text{data}}$, by mapping $\mathbf{x}_T$ back through a continuous transformation process. Let $\{\mathbf{x}_s\}_{s=0}^T$ represent a continuous transformation trajectory from $\mathbf{x}_T$ to $\mathbf{x}_0$, where $\mathbf{x}_s$ denotes the state at intermediate time $s \in [0, T]$. To describe the reverse-time dynamics that map $\mathbf{x}_T$ back to $\mathbf{x}_0$, we define a reverse-time ODE:

$$\frac{d\mathbf{x}_s}{ds} = \mathbf{v}(\mathbf{x}_s, s), \tag{1}$$

where $\mathbf{v} : \mathbb{R}^n \times [0, T] \to \mathbb{R}^n$ is a continuous function defining the system's dynamics in reverse time.

The process of obtaining $\mathbf{x}_0$ from $\mathbf{x}_T$ involves solving this reverse-time ODE, which can be understood as computing the integral:

$$\int_T^0 \frac{\mathrm{d}\mathbf{x}_s}{\mathrm{d}s}\,\mathrm{d}s = \int_T^0 \mathbf{v}(\mathbf{x}_s, s)\,\mathrm{d}s \iff \mathbf{x}_0 = \mathbf{x}_T + \int_T^0 \mathbf{v}(\mathbf{x}_s, s)\,\mathrm{d}s \tag{2}$$

The solution of the reverse-time ODE aligns marginally in distribution with the forward process, meaning that the distribution of $\mathbf{x}_0$ obtained by solving the reverse ODE starting from $\mathbf{x}_T \sim p_T(\mathbf{x})$ approximates the target distribution $p_{data}$.

While traditional ODE solvers and neural ODE methods are commonly used to solve the equation 2 they come with notable drawbacks. The numerical solvers of ODE can not avoid the discretization error (Bortoli et al., 2023) , which restricts the quality of samples when only a few NFEs are used. Second, neural ODEs (Chen et al., 2018), which approximate the ODE solution using neural networks, faces a high challenge during gradient backpropagation due to their high memory consumption (Gholami et al., 2019).

### 3.2 INTEGRATION FLOWS

To overcome the challenges associated with ODE solvers and neural ODEs, we propose Integration Flow to directly estimate the integrated effect of continuous-time dynamics. Integration Flow explicitly incorporating the target state $\mathbf{x}_0$ as the anchor state in guiding the reverse-time dynamics from the intermediate state $\mathbf{x}_t$, which contributes to both stability and accuracy in reconstructing $\mathbf{x}_0$ from intermediate states $\mathbf{x}_t$. Since Integration Flow bypasses the ODE solver, it provides a unified framework for ODE-based generative models, allowing for one-step generation across a variety of processes.

The cumulative effect of the reverse-time dynamics over the interval $[0, t]$, which is defined in integral $\int_t^0 \mathbf{v}(\mathbf{x}_s, s)ds$, can be obtained as:

$$\int_t^0 \mathbf{v}(\mathbf{x}_s, s)ds = \mathbf{V}(\mathbf{x}_0, 0) - \mathbf{V}(\mathbf{x}_t, t)$$

where $\mathbf{V}(\mathbf{x}_s, s)$ is an antiderivative of $\mathbf{v}(\mathbf{x}_s, s)$ with respect to $s$. Then, we defined a function $\mathbf{F}(\mathbf{x}_0, \mathbf{x}_t, t)$ as follows:

$$\mathbf{F}(\mathbf{x}_0, \mathbf{x}_t, t) := \mathbf{V}(\mathbf{x}_t, t) - \mathbf{V}(\mathbf{x}_0, 0)$$

This function encapsulates the total influence of the dynamics from an intermediate time $t$ to the final time 0, which leads to the equation:

$$\mathbf{x}_0 = \mathbf{x}_t - \mathbf{F}(\mathbf{x}_0, \mathbf{x}_t, t).$$

Next, we define the function:

$$\mathbf{f}(\mathbf{x}_0, \mathbf{x}_t, t) := \mathbf{x}_t - \mathbf{F}(\mathbf{x}_0, \mathbf{x}_t, t). \tag{3}$$

Therefore, we have:

$$\mathbf{x}_0 = \mathbf{f}(\mathbf{x}_0, \mathbf{x}_t, t). \tag{4}$$

Thus, $\mathbf{f}(\mathbf{x}_0, \mathbf{x}_t, t)$ is the **solution** of the reversed time ODE from initial time $t$ to final time $0$, which encapsulates the cumulative effect of the reverse dynamics from the initial time $t$ to the final time $0$, providing an accumulation description of how the the target state $\mathbf{x}_0$ transformed from the intermediate state $\mathbf{x}_t$. The inclusion of $\mathbf{x}_0$ in $\mathbf{f}(\mathbf{x}_0, \mathbf{x}_t, t)$ helps stabilize the generative process by incorporating information about the final state, leading to improved accuracy in reconstructing the intermediate states while maintaining consistency with the target distribution.

### 3.3 NEURAL NETWORK APPROXIMATION

In practice, the exact form of $\mathbf{F}(\mathbf{x}_0, \mathbf{x}_t, t)$ is usually intractable or unknown. Therefore, we model $\mathbf{F}$ using a neural network parameterized by $\boldsymbol{\theta}$. The approximated predictive model is thus defined as:

$$\mathbf{f}_{\boldsymbol{\theta}}(\mathbf{x}_0, \mathbf{x}_t, t) = \mathbf{x}_t - \mathbf{F}_{\boldsymbol{\theta}}(\mathbf{x}_0, \mathbf{x}_t, t) \tag{5}$$

where $\mathbf{F}_{\boldsymbol{\theta}}$ approximates $\mathbf{F}$.

To improve performance, especially in complex scenarios such as VE case of diffusion model or the PFGM++ model (will be shown in section 4), a more robust and flexible formulation is required to ensure the stability of the integration flow. We redefine the dynamics $\mathbf{f}_{\boldsymbol{\theta}}(\mathbf{x}_0, \mathbf{x}_t, t)$ as the following:

$$\mathbf{f}_{\boldsymbol{\theta}}(\mathbf{x}_0, \mathbf{x}_t, t) = a_t \mathbf{x}_t + b_t \mathbf{F}_{\boldsymbol{\theta}}(\mathbf{x}_0, \mathbf{x}_t, t). \tag{6}$$

where $a_t$ and $b_t$ are time-dependent scalar functions designed to modulate the contributions of $\mathbf{x}_t$ and $\mathbf{F}(\mathbf{x}_0, \mathbf{x}_t, t)$, respectively. This formulation introduces greater flexibility in the evolution of the integration flow over time, particularly in scenarios where the straightforward application (equation 5) of integration may introduce instability, especially when the magnitudes of the intermediate state $\mathbf{x}_t$ become large.

To achieve accurate recovery of $\mathbf{x}_0$, it is essential that:

$$\mathbf{f}_{\boldsymbol{\theta}}(\mathbf{x}_0, \mathbf{x}_t, t) \approx \mathbf{f}(\mathbf{x}_0, \mathbf{x}_t, t)$$

Recovering $\mathbf{x}_0$ from $\mathbf{x}_t$ is achieved through an iterative refinement process. Starting with an initial $\mathbf{x}_0^{(0)}$, the estimate is progressively refined using the update rule:

$$\mathbf{x}_0^{(n+1)} = \mathbf{f}_{\boldsymbol{\theta}}\left(\mathbf{x}_0^{(n)}, \mathbf{x}_t, t\right) = a_t \mathbf{x}_t + b_t \mathbf{F}_{\boldsymbol{\theta}}\left(\mathbf{x}_0^{(n)}, \mathbf{x}_t, t\right)$$

Through this iterative process and proper defined loss, the neural network will effectively minimize the discrepancy between the iteratively estimated $\mathbf{x}_0^{(n)}$ and the true initial state $\mathbf{x}_0$.

### 3.4 THEORETICAL JUSTIFICATION

**Theorem 1 (Stability)**: Let $\mathbf{x}_0$ represent the target state, $\mathbf{x}_t$ represent an intermediate state, and $t$ represent the time. Let $\mathbf{x}_0^{(n)}$ be an auxiliary estimate of $\mathbf{x}_0$ obtained through an iterative process. Consider the following two estimators: (a) $\hat{\mathbf{x}}_0 = \mathbf{f}_{\boldsymbol{\theta}}(\mathbf{x}_t, t)$, which estimates $\mathbf{x}_0$ based only on $\mathbf{x}_t$ and $t$, analogous to $\mathbb{E}[\mathbf{x}_0|\mathbf{x}_t]$. (b) $\tilde{\mathbf{x}}_0 = \mathbf{f}_{\boldsymbol{\theta}}\left(\mathbf{x}_0^{(n)}, \mathbf{x}_t, t\right)$, which estimates $\mathbf{x}_0$ based on both $\mathbf{x}_0^{(n)}, \mathbf{x}_t$, and $t$, analogous to $\mathbb{E}[\mathbf{x}_0|\mathbf{x}_t, \mathbf{x}_0^{(n)}]$. Then, the estimator $\tilde{\mathbf{x}}_0$, which includes additional conditional information $\mathbf{x}_0^{(n)}$, provides a more accurate estimation of $\mathbf{x}_0$ compared to $\hat{\mathbf{x}}_0$, in terms of mean squared error (MSE). That is,

$$\mathbb{E}\left[\|\mathbf{x}_0 - \tilde{\mathbf{x}}_0\|^2\right] \leq \mathbb{E}\left[\|\mathbf{x}_0 - \hat{\mathbf{x}}_0\|^2\right]$$

Moreover, it can be expand to any convex metric $d(\cdot, \cdot)$. That is,

$$\mathbb{E}[d(\mathbf{x}_0, \tilde{\mathbf{x}}_0)] \leq \mathbb{E}[d(\mathbf{x}_0, \hat{\mathbf{x}}_0)].$$

Table 1: The different design choice of Integration Flow for different ODE-based methods. For training, we use the discrete time steps with $T = 1000$.

| | VE(Song et al., 2020b) | Rectified Flow(Liu et al., 2022) | PFGM++Xu et al. (2023) |
|---|---|---|---|
| **Training** | | | |
| Noise scheduler | $\sigma_{\min}\left(\frac{\sigma_{\max}}{\sigma_{\min}}\right)^{t/T}$ | Linear interpolation, $\sigma = 0$ | $R_t\mathbf{v}_t$, where $\sigma_t \sim p(\sigma_t)$, $r_t = \sigma_t\sqrt{D}$, $R_t \sim p_{r_t}(R)$, $\mathbf{v}_t = \frac{\mathbf{u}_t}{\|\mathbf{u}_t\|_2}, \mathbf{u}_t \sim \mathcal{N}(\mathbf{0}, \mathbf{I})$ |
| Steps | $t \in [1, 2, ..., T]$ | $t \sim \text{Uniform}[0, 1]$ | $t \in [1, 2, ..., T]$ |
| **Network and preconditioning** | | | |
| Architecture of $F_\theta$ | ADM | ADM | ADM |
| $a_t$ | $\sigma_{\min}/\sigma_t$ | 0 | $\sigma_{\min}/R_t$ |
| $b_t$ | $1 - \sigma_{\min}/\sigma_t$ | 1 | $1 - \sigma_{\min}/R_t$ |
| **Sampling** | | | |
| One step | | $\mathbf{x}_0^{(\text{est})} = a_t\mathbf{x}_t + b_t\mathbf{F}_{\boldsymbol{\theta}}\left(\mathbf{x}_0^{(0)}, \mathbf{x}_t, t\right)$ | |
| Multistep $n$ | | $\mathbf{x}_0^{(\text{est})} = a_t\mathbf{x}_t + b_t\mathbf{F}_{\boldsymbol{\theta}}\left(\mathbf{x}_0^{(n)}, \mathbf{x}_t, t\right)$ | |
| **Parameters** | | | |
| | $\sigma_{\min} = 0.01$ | | $\sigma_{\min} = 0.01$ |
| | $\sigma_{\max} = 50$ | —— | $\sigma_{\max} = 50$ |
| | | | $D = 2048$ |

Theorem 1 justifies that the estimator $\tilde{\mathbf{x}}_0$ is at least as accurate as $\hat{\mathbf{x}}_0$ under the same convex metric $d(\cdot, \cdot)$, illustrating that $\tilde{\mathbf{x}}_0 = \mathbf{f}_{\boldsymbol{\theta}}\left(\mathbf{x}_0^{(n)}, \mathbf{x}_t, t\right)$ provides a better or at least equal estimation compared to $\hat{\mathbf{x}}_0 = \mathbf{f}_{\boldsymbol{\theta}}\left(\mathbf{x}_t, t\right)$.

**Theorem 2 (Non-Intersection)**: Suppose the neural network is sufficiently trained and $\boldsymbol{\theta}^*$ is obtained, such that: $\mathbf{f}_{\boldsymbol{\theta}^*}(\mathbf{x}_0^{(n)}, \mathbf{x}_t, t) \equiv \mathbf{f}(\mathbf{x}_0, \mathbf{x}_t, t)$ for any $t \in [0, T]$ and $\mathbf{x}_0$ sampled from $p_{\text{data}}$, and $\mathbf{v}(\mathbf{x}_s, s)$ meets Lipschitz condition.

Then for any $t \in [0, T]$, the mapping $\mathbf{f}_{\boldsymbol{\theta}^*}(\mathbf{x}_0^{(n)}, \mathbf{x}_t, t) : \mathbb{R}^N \to \mathbb{R}^N$ is bi-Lipschitz. Namely, for any $\mathbf{x}_t, \mathbf{y}_t \in \mathbb{R}^N$

$$e^{-Lt}\|\mathbf{x}_t - \mathbf{y}_t\|_2 \leq \left\|\mathbf{f}_{\boldsymbol{\theta}^*}(\mathbf{x}_0^{(n)}, \mathbf{x}_t, t) - \mathbf{f}_{\boldsymbol{\theta}^*}(\mathbf{y}_0^{(n)}, \mathbf{y}_t, t)\right\|_2 \leq e^{Lt}\|\mathbf{x}_t - \mathbf{y}_t\|_2.$$

This implies that if given two different starting point, say $\mathbf{x}_T \neq \mathbf{y}_T$, by the bi-Lipschitz above, it can be conculde that $\mathbf{f}_{\boldsymbol{\theta}^*}(\mathbf{x}_0^{(n)}, \mathbf{x}_T, T) \neq \mathbf{f}_{\boldsymbol{\theta}^*}(\mathbf{y}_0^{(n)}, \mathbf{y}_T, T)$ i.e., $\mathbf{x}_0^{(n+1)} \neq \mathbf{y}_0^{(n+1)}$, which indicate the reverse path of Integration Flow does not intersect.

The proof of Theorems presented in Appendix B.

# 4 INTEGRATION FLOW FOR DIFFERENT ODE-BASED GENERATIVE MODELS

In this section, we explain how Integration Flow can be applied to three ODE-based generative models. More training settings can be seen in Table 1.

**VE case of diffusion models**: For the intermediate state $\mathbf{x}_t = \mathbf{x}_0 + \sigma_t\epsilon$, We adopt the noise scheduler as $\sigma_{\min}\left(\frac{\sigma_{\max}}{\sigma_{\min}}\right)^{t/T}$, where noise increases exponentially over time from $\sigma_{\min}$ to $\sigma_{\max}$, and time step $t$ is designed as $t \in [1, 2, ..., T]$.

The integration flow can be expressed as:

$$\mathbf{f}_{\boldsymbol{\theta}}(\mathbf{x}_0, \mathbf{x}_t, t) = \frac{\sigma_{\min}}{\sigma_t}\mathbf{x}_t + \left(1 - \frac{\sigma_{\min}}{\sigma_t}\right)F_{\boldsymbol{\theta}}(\mathbf{x}_0, \mathbf{x}_t, t)$$

where the preconditioning terms are set as $a_t = \sigma_{\min}/\sigma_t$, and $b_t = 1 - \sigma_{\min}/\sigma_t$, which modulate the network's response to different noise levels throughout training. The detailed derivation of Integration Flow for VE diffusion model is in A.1.

**Rectified Flows**: the intermediate is expressed as: $\mathbf{z}_t = (1 - t)\mathbf{x}_0 + t\mathbf{z}$, time step $t$ is sampled from Uniform$[0, 1]$. Since this is a deterministic linear interpolation, so there is no need of noise scheduler.

The integration flow of Rectified Flows can be expressed as:

$$\mathbf{x}_0 = \mathbf{f}_{\boldsymbol{\theta}}(\mathbf{x}_0, \mathbf{z}_t, t) = F_{\boldsymbol{\theta}}(\mathbf{x}_0, \mathbf{z}_t, t)$$

Equivalent to $a_t = 0, b_t = 1$ in equation 6. The detailed derivation of Integration Flow for Rectified Flow is in A.2. Moreover, Integration Flow supports Stochastic Interpolants as well.

**PFGM++**: PFGM++ introduces an alignment method to transfer hyperparameters from diffusion models (where $D \to \infty$) to finite-dimensional settings. The alignment is based on the relationship:

$$r = \sigma\sqrt{D}$$

This formula ensures that the phases of the intermediate distributions in PFGM++ are aligned with those of diffusion models. The relation allows transferring finely-tuned hyperparameters like $\sigma_{\max}$ and $p(\sigma)$ from diffusion models to PFGM++ using:

$$r_{\max} = \sigma_{\max}\sqrt{D}, \quad p(r) = \frac{p(\sigma = r/\sqrt{D})}{\sqrt{D}}$$

Further, (Xu et al., 2023) showed

$$\frac{d\mathbf{x}}{dr} = \frac{d\mathbf{x}}{d\sigma}$$

where $\sigma$ changes with time. Thus, we adopt the noise scheduler same as in VE case. And the perturbation to the original data $\mathbf{x}_0$ can be written as:

$$\mathbf{x}_t = \mathbf{x}_0 + R_t\mathbf{v}_t$$

Specifically, for each data point $\mathbf{x}_0$, a radius $R_t$ is sampled from the distribution $p_{r_t}(R)$(See Appendix B in (Xu et al., 2023) to sample $R_t$). To introduce random perturbations, uniform angles are sampled by first drawing from a standard multivariate Gaussian, $\mathbf{u}_t \sim \mathcal{N}(\mathbf{0}, \mathbf{I})$, and then normalizing these vectors to obtain unit direction vectors $\mathbf{v}_t = \frac{\mathbf{u}_t}{\|\mathbf{u}_t\|_2}$. This perturbation acts as a forward process in PFGM++, analogous to the forward process in diffusion models.

The Integration Flow $\mathbf{f}_{\boldsymbol{\theta}}(\mathbf{x}_0, \mathbf{x}_t, \sigma_t)$ of PFGM++ can be expressed as

$$\mathbf{f}_{\boldsymbol{\theta}}(\mathbf{x}_0, \mathbf{x}_t, t) = \frac{\sigma_{\min}}{R_t}\mathbf{x}_t + (1 - \frac{\sigma_{\min}}{R_t})\mathbf{F}_{\boldsymbol{\theta}}(\mathbf{x}_0, \mathbf{x}_t, t)$$

with $a_t = \sigma_{\min}/R_t$ and $b_t = 1 - \sigma_{\min}/R_t$, and the detailed derivation of $a_t, b_t$ is shown in Appendix A.3.

## 5 EXPERIMENTS

To evaluate our method for image generation, we train several Integration Flow Models on CIFAR-10 Krizhevsky et al. (2009) and benchmark their performance with competing methods in the literature. Results are compared according to Frechet Inception Distance (FID, Heusel et al. (2017)), which is computed between 50K generated samples and the whole training set. The training and sampling algorithm can be found in Appendix A.

### 5.1 IMPLEMENTATION DETAILS

**Architecture.** We use the U-Net architecture from ADM Dhariwal & Nichol (2021) for the dataset. For CIFAR-10, we use a base channel dimension of 128, multiplied by 1,2,2,2 in 4 stages and 3 residual blocks per stage. Dropout Srivastava et al. (2014) of 0.3 is utilized for this task. Following ADM, we employ cross-attention modules not only at the 16x16 resolution but also at the 8x8 resolution, through which we incorporate the conditioning image $\mathbf{x}_0^{(n)}$ into the network.We also explore deeper variants of these architectures by doubling the number of blocks at each resolution, which we name Integration Flow-deep. All models on CIFAR-10 are unconditional.

Table 2: Comparing the quality of unconditional samples on CIFAR-10

| Method | NFE(↓) | FID(↓) | IS(↑) |
|---|---|---|---|
| **Fast samplers & distillation for diffusion models** | | | |
| DDIM Song et al. (2020a) | 10 | 13.36 | |
| DPM-solver-fast Lu et al. (2022) | 10 | 4.70 | |
| 3-DEIS Zhang & Chen (2022) | 10 | 4.17 | |
| UniPC Zhao et al. (2023) | 10 | 3.87 | |
| DFNO (LPIPS) Zheng et al. (2023) | 1 | 3.78 | |
| 2-Rectified Flow Liu et al. (2022) | 1 | 4.85 | 9.01 |
| Knowledge Distillation Luhman & Luhman (2021) | 1 | 9.36 | |
| TRACT Berthelot et al. (2023) | 1 | 3.78 | |
| | 2 | 3.32 | |
| Diff-Instruct Luo et al. (2023) | 1 | 4.53 | 9.89 |
| CD (LPIPS) Song et al. (2023) | 1 | 3.55 | 9.48 |
| | 2 | 2.93 | 9.75 |
| **Direct Generation** | | | |
| Score SDE Song et al. (2020b) | 2000 | 2.38 | 9.83 |
| Score SDE (deep) Song et al. (2020b) | 2000 | 2.20 | 9.89 |
| DDPM (Ho et al., 2020) | 1000 | 3.17 | 9.46 |
| LSGM Vahdat et al. (2021) | 147 | 2.10 | |
| PFGM Xu et al. (2022) | 110 | 2.35 | 9.68 |
| EDM Karras et al. (2022) | 35 | 2.04 | 9.84 |
| PFGM++ (D=2048) Xu et al. (2023) | 35 | 1.91 | 9.68 |
| EDM-G++ Kim et al. (2022) | 35 | 1.77 | |
| 1-Rectified Flow(Liu et al., 2022) | 1 | 378 | 1.13 |
| NVAE Vahdat & Kautz (2020) | 1 | 23.5 | 7.18 |
| BigGAN Brock et al. (2018) | 1 | 14.7 | 9.22 |
| StyleGAN2 Karras et al. (2020a) | 1 | 8.32 | 9.21 |
| StyleGAN2-ADA Karras et al. (2020b) | 1 | 2.92 | 9.83 |
| CT (LPIPS) Song et al. (2023) | 1 | 8.70 | 8.49 |
| | 2 | 5.83 | 8.85 |
| iCT Song & Dhariwal (2023) | 1 | 2.83 | 9.54 |
| | 2 | 2.46 | 9.80 |
| iCT-deep Song & Dhariwal (2023) | 1 | 2.51 | 9.76 |
| | 2 | 2.24 | 9.89 |
| **Integration Flow(VE)** | 1 | 2.87 | 9.56 |
| | 2 | 2.64 | 9.76 |
| | 1000 | 1.89 | 9.93 |
| **Integration Flow (VE-deep)** | 1 | 2.63 | 9.77 |
| | 2 | 2.35 | 9.88 |
| | 1000 | 1.71 | 9.95 |
| **Integration Flow (1-Rectified Flow)** | 1 | **3.40** | 9.48 |
| **Integration Flow (PFGM++, D=2048)** | 1 | **2.96** | |

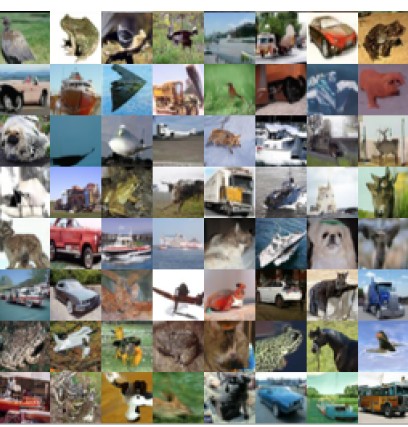

Figure 2: One-step samples from Integration Flow-VE

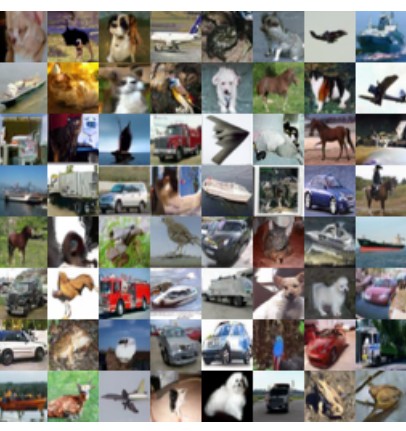

Figure 3: One-step samples from Integration Flow-VE-deep

**Loss function** Inspired by Song & Dhariwal (2023), we adopt the Pseudo-Huber metric family Charbonnier et al. (1997) as the loss function, defined as

$$d(\boldsymbol{x}, \boldsymbol{y}) = \sqrt{\|\boldsymbol{x} - \boldsymbol{y}\|_2^2 + c^2} - c \qquad (7)$$

where $c$ is an adjustable hyperparameter. The Pseudo-Huber metric is more robust to outliers compared to the squared $\ell_2$ loss metric because it imposes a smaller penalty for large errors, while still behaving similarly to the squared $\ell_2$ loss metric for smaller errors. We set $c = 0.00015$ for VE, $c = 0.00014$ for Rectified Flow and $c = 0.00014$ for PFGM++, respectively.

**Other settings.** We use Adam for all of our experiments. For CIFAR-10, we set $T = 1000$ for baseline model and train the model for 400,000 iterations with a constant learning rate of 0.0002 and batch size of 1024. We use an exponential moving average (EMA) of the weights during training with a decay factor of 0.9999 for all the experiments. All models are trained on 8 Nvidia H100 GPUs.

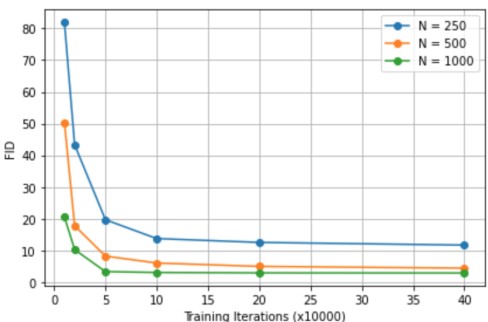

Figure 4: FID Values vs. Training Iterations for Different $N$.

## 5.2 FAST CONVERGENCE OF INTEGRATION FLOW

As shown on Figure 4, the Integration Flow converge fast compared with the (Song & Dhariwal, 2023), especially for the larger values of N (500 and 1000). The steep decline in FID scores during the early iterations (particularly from 10000 to 100000 iterations) indicates that the models are learning quickly and that performance stabilizes after a relatively small number of iterations.

## 5.3 COMPARISON TO SOTA

We compare our model against state-of-the-art generative models on CIFAR-10. Quantitative results are summarized in Table 2. Our findings reveal that Integration Flow exceed previous distillation diffusion models and methods that require advanced sampling procedures in both one-step and two-step generation, which breaks the reliance on the well-pretrained diffusion models and simplifies the generation workflow. Moreover, our model demonstrates performance comparable to numerous leading generative models for VE settings. Specifically, baseline Integration Flow obtains FIDs of 2.87 for one-step generation in VE, results exceed that of StyleGAN2-ADA (Karras et al., 2020b). For deeper architecture, our model achieves one-step generation with FID of 2.63 for VE. Additionally, VP-deep outperforms the leading model iCT-deep (Song & Dhariwal, 2023) on two-step generation. With 1000-step sampling, VE-deep push FID to 1.71, setting state-of-the-art performance in both cases.

For Rectified Flow, the one-step generation with Integration Flow has reached 3.4 for FID without reflow, which is also the state-of-the-art performance in the Rectified Flow. Generally, the Rectified Flow need to be applied at least twice (reflow) to obtain a reasonable on-step generation performance(Liu et al., 2022; 2023). For PFGM++, Integration Flow has reached 2.96 of FID in one-step generation. We are the first to show that the PFGM++ can also achieve one-step generation with good performance.

## 6 DISCUSSION

Integration Flow presents a straightforward yet powerful approach that unifies different types of ODE-based generative models. Its core strength is simplicity: instead of relying on complex iterative sampling or solving ODEs step-by-step, Integration Flow directly learns the overall transformation dynamics, allowing for one-step generation. By using a pseudo-Huber loss function—simple and easy to work with—the model benefits from stable training and minimal parameter tuning, making it both scalable and adaptable across various ODE-based frameworks.

A key achievement of Integration Flow is its ability to solve different ODE-based models using a single framework, addressing a significant challenge that prior models struggled with. Fore example, two important generative models, Rectified Flow and diffusion model, are not unified, but Integration Flow can successfully integrate them. This not only simplifies the landscape of ODE-based generative models but also expands their applicability, making them easier to implement across different domains.

Since Integration Flow keeps track of $\mathbf{x}_0^{(n)}$ for each sample in the dataset, there will be additional memory consumption during training. Specifically, it requires extra 614MB for CIFAR-10. Although it can be halved by using FP16 data type, such memory requirement might still be a challenge for larger dataset or dataset with high-resolution images. One solution is to store $\mathbf{x}_0^{(n)}$ in a buffer or on disk instead of on the GPU. However, this approach will introduce additional overhead during training due to the need to transfer data back to the GPU. We will fix this out as our future work.

Although, Integration Flow has achieved best performance on one-step generation for Rectified Flow and PFGM++, the performance of Integration Flow for VE is still slightly underperformed compared to current the-state-of-the-art. We hypothesize that this small performance gap may be attributed to suboptimal hyperparameters in the loss function. Due to limited computation resouce, we are not able to search the best hyperparameter. Additionally, we recognize that different noise schedulers can significantly impact the model's performance. The noise scheduling strategy plays a crucial role in the training dynamics and final performance of the model. We plan to investigate more complex schedulers in future work.

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

# A PRECONDITIONING SETTINGS AND ALGORITHMS

In this Appendix, we give detailed derivation of Integration Flow for the VE case of Diffusion Model, Rectified Flow and PFGM++.

## A.1 INTEGRATION FLOW FOR VE DIFFUSION MODELS

The PF-ODE of VE diffusion models is formulated as:

$$\frac{d\mathbf{x}_t}{dt} = -\frac{1}{2}\frac{d\sigma_t^2}{dt}\nabla_{\mathbf{x}_t}\log p_t\left(\mathbf{x}_t\right)$$

We do the reversed time integration on both sides over the interval $[0, t]$:

$$\int_t^0 \frac{d\mathbf{x}_s}{ds} = \int_t^0 -\frac{1}{2}\frac{d\sigma_t^2}{dt}\nabla_{\mathbf{x}_t}\log p_t\left(\mathbf{x}_t\right)$$

and obtain:

$$\mathbf{x}_0 - \mathbf{x}_t = \mathbf{V}\left(\mathbf{x}_0, 0\right) - \mathbf{V}\left(\mathbf{x}_t, t\right) = -F\left(\mathbf{x}_0, \mathbf{x}_t, t\right)$$

Thus:

$$\mathbf{x}_0 = \mathbf{x}_t - F\left(\mathbf{x}_0, \mathbf{x}_t, t\right) = \mathbf{f}\left(\mathbf{x}_0, \mathbf{x}_t, t\right)$$

For stable training purpose, we rewrite $\mathbf{f}\left(\mathbf{x}_0, \mathbf{x}_t, t\right)$ as following:

$$\mathbf{f}\left(\mathbf{x}_0, \mathbf{x}_t, t\right) = \mathbf{x}_t - F\left(\mathbf{x}_0, \mathbf{x}_t, t\right)$$
$$= \kappa\left(\sigma_t\right)\mathbf{x}_t + \left(1 - \kappa\left(\sigma_t\right)\right)\left[\mathbf{x}_t - \frac{1}{1 - \kappa\left(\sigma_t\right)}F\left(\mathbf{x}_0, \mathbf{x}_t, t\right)\right]$$

We define the neural network as:

$$\mathbf{f}_{\boldsymbol{\theta}}\left(\mathbf{x}_0, \mathbf{x}_t, t\right) = \kappa\left(\sigma_t\right)\mathbf{x}_t + \frac{1}{1 - \kappa\left(\sigma_t\right)}F_{\boldsymbol{\theta}}\left(\mathbf{x}_0, \mathbf{x}_t, t\right),$$

where we use neural network to estimate the value of $\mathbf{x}_t - \left(1 - \kappa\left(\sigma_t\right)\right)F\left(\mathbf{x}_0, \mathbf{x}_t, t\right)$.

For VE Diffusion model, we have $a(\sigma_t) = \kappa(\sigma_t)$ and $b(\sigma_t) = 1 - \kappa(\sigma_t)$. There are a few choices for the design of $\kappa(\sigma_t)$, such as $\kappa(\sigma_t) = \frac{\sigma_{\text{data}}}{\sigma_t + \sigma_{\text{data}}}$, $\kappa(\sigma_t) = \frac{\sigma_{\text{data}}^2}{\sigma_t^2 + \sigma_{\text{data}}^2}$, which are in a manner of (Karras et al., 2022). We set $\kappa(\sigma_t) = \frac{\sigma_{\min}}{\sigma_t}$ in this work.

---

**Algorithm 1** Integration Flow Training Algorithm for VE Diffusion Model

---

**Input:** $p_{\text{data}}, T$, model parameter $\boldsymbol{\theta}$, initialize $\mathbf{x}_0^{(0)} \sim \mathcal{N}(\mathbf{0}, \mathbf{I})$, epoch $n \leftarrow 0$
**repeat**
    Sample $\mathbf{x}_0 \sim p_{\text{data}}, t \sim \mathcal{U}\left[1, T\right]$ and $\boldsymbol{\epsilon} \sim \mathcal{N}(\mathbf{0}, \mathbf{I})$
    $\mathbf{x}_t = \mathbf{x}_0 + \sigma_t\boldsymbol{\epsilon}$
    $\mathbf{x}_0^{(n+1)} \leftarrow \mathbf{f}_{\boldsymbol{\theta}}\left(\mathbf{x}_0^{(n)}, \mathbf{x}_t, t\right)$
    $\mathcal{L}_{\text{IFM}}^{(n+1)}(\boldsymbol{\theta}) \leftarrow d\left(\mathbf{f}_{\boldsymbol{\theta}}\left(\mathbf{x}_0^{(n)}, \mathbf{x}_t, t\right), \mathbf{x}_0\right)$
    $\boldsymbol{\theta} \leftarrow \boldsymbol{\theta} - \eta\nabla_{\boldsymbol{\theta}}\mathcal{L}(\boldsymbol{\theta})$
    $n \leftarrow n + 1$
**until** convergence

---

---

**Algorithm 2** Integration Flow Sampling Algorithm for VE Diffusion Mode

---

**Input:** $T$, trained model parameter $\boldsymbol{\theta}$, sampling step $k$, initialize $\mathbf{x}_0^{(0)} \sim \mathcal{N}(\mathbf{0}, \mathbf{I})$, $\mathbf{x}_T \sim \mathcal{N}(\mathbf{0}, \mathbf{I})$
$\mathbf{x}_T = \sigma_{\max} \mathbf{x}_T$
**for** $k = 0$ **to** $k - 1$ **do**
$\quad \mathbf{x}_0^{(k+1)} \leftarrow \mathbf{f}_{\boldsymbol{\theta}} \left( \mathbf{x}_0^{(k)}, \mathbf{x}_T, T \right)$
**end for**
**Output:** $\mathbf{x}_0^{(k+1)}$

---

### A.2   INTEGRATION FLOW FOR RECTIFIED FLOW

To analyze the integration flow associated with this process, we consider the derivative of $\mathbf{z}_t$ with respect to time $t \in [0, 1]$:

$$\frac{d\mathbf{z}_t}{dt} = \mathbf{v}\left(\mathbf{z}_t, t\right)$$

Integrating the both side, we obtain:

$$\int_0^1 \frac{d\mathbf{z}_s}{ds} ds = \mathbf{z} - \mathbf{x}_0.$$

This confirms that the total change over the entire path from $t = 0$ to $t = 1$ is simply the difference between the endpoints $\mathbf{z}_1$ and $\mathbf{x}_0$.

For any intermediate time $t \in [0, 1]$, reserve-time integration over $[0, t]$ yields:

$$\int_t^0 \frac{d\mathbf{z}_s}{ds} ds = \mathbf{x}_0 - \mathbf{z}_t = \mathbf{V}\left(\mathbf{x}_0, 0\right) - \mathbf{V}\left(\mathbf{x}_t, t\right) = -F(\mathbf{x}_0, \mathbf{z}_t, t),$$

where we define the accumulated change $F(\mathbf{x}_0, \mathbf{z}_t, t)$ as:

$$F(\mathbf{x}_0, \mathbf{z}_t, t) = \mathbf{z}_t - \mathbf{x}_0.$$

Substituting the expression for $\mathbf{z}_t$, we have:

$$\mathbf{z}_t - \mathbf{x}_0 = [(1 - t)\mathbf{x}_0 + t\mathbf{z}] - \mathbf{x}_0 = t(\mathbf{z} - \mathbf{x}_0).$$

Thus, the accumulated change is proportional to the time parameter $t$ and the difference $\mathbf{z} - \mathbf{x}_0$:

$$F(\mathbf{x}_0, \mathbf{z}_t, t) = t(\mathbf{z} - \mathbf{x}_0).$$

Rearranging this expression allows us to solve for $\mathbf{z} - \mathbf{x}_0$:

$$\mathbf{z} - \mathbf{x}_0 = \frac{F(\mathbf{x}_0, \mathbf{z}_t, t)}{t}.$$

This relationship indicates that the total accumulated change $\mathbf{z} - \mathbf{x}_0$ can be expressed in terms of the accumulated change $F(\mathbf{x}_0, \mathbf{z}_t, t)$ scaled by $1/t$.

We can now define the Integration Flow of the Rectified Flow process by expressing $\mathbf{x}_0$ in terms of $F(\mathbf{x}_0, \mathbf{z}_t, t)$ and the endpoint $\mathbf{z}$:

$$\mathbf{x}_0 = \mathbf{z} - \frac{F(\mathbf{x}_0, \mathbf{z}_t, t)}{t} = \mathbf{f}\left(\mathbf{x}_0, \mathbf{x}_t, t\right).$$

In practice, since $\mathbf{z}$ is deterministic, it can be absorbed into the neural network; for stable training, we take $F(\mathbf{x}_0, \mathbf{z}_t, t)/t$ as a whole.

Thus ,we have:

$$\mathbf{f}_{\boldsymbol{\theta}}(\mathbf{x}_0, \mathbf{x}_t, t) = F_{\boldsymbol{\theta}}(\mathbf{x}_0, \mathbf{x}_t, t),$$

which indicates $a_t = 0$ and $b_t = 1$.

By employing this enhanced dynamic model within the Rectified Flow framework, we can achieve a more accurate and stable reconstruction of the initial data point $\mathbf{x}_0$, facilitating effective generative modeling and data synthesis.

---

**Algorithm 3** Integration Flow Training Algorithm for Rectified Flows

---

**Input:** couple $(\mathbf{x}_0, \mathbf{z})$ from $p_{\text{data}}$ and $p_{\mathbf{z}}$, respectively; model parameter $\boldsymbol{\theta}$, initialize $\mathbf{x}_0^{(0)} \sim \mathcal{N}(\mathbf{0}, \mathbf{I})$, epoch $n \leftarrow 0$
**repeat**
    Sample $\mathbf{x}_0 \sim p_{\text{data}}, t \sim \text{Uniform}[0, 1]$
    $\mathbf{z}_t = (1 - t)\mathbf{x}_0 + t\mathbf{z}$
    $\mathbf{x}_0^{(n+1)} \leftarrow \mathbf{f}_{\boldsymbol{\theta}}\left(\mathbf{x}_0^{(n)}, \mathbf{z}_t, t\right)$
    $\mathcal{L}_{\text{IFM}}^{(n+1)}(\boldsymbol{\theta}) \leftarrow d\left(\boldsymbol{f}_{\boldsymbol{\theta}}\left(\mathbf{x}_0^{(n)}, \mathbf{z}_t, t\right), \mathbf{x}_0\right)$
    $\boldsymbol{\theta} \leftarrow \boldsymbol{\theta} - \eta \nabla_{\boldsymbol{\theta}} \mathcal{L}(\boldsymbol{\theta})$
    $n \leftarrow n + 1$
**until** convergence

---

---

**Algorithm 4** Integration Flow Sampling Algorithm for Rectified Flows

---

**Input:** $t = 1$, trained model parameter $\boldsymbol{\theta}$, draw $\mathbf{z} \sim p_{\mathbf{z}}$, initialize $\mathbf{x}_0^{(0)} \sim \mathcal{N}(\mathbf{0}, \mathbf{I})$, sampling step $k$, initialize $\mathbf{x}_0^{(0)} \sim \mathcal{N}(\mathbf{0}, \mathbf{I})$
**for** $k = 0$ **to** $k - 1$ **do**
    $\mathbf{x}_0^{(k+1)} \leftarrow \mathbf{f}_{\boldsymbol{\theta}}\left(\mathbf{x}_0^{(k)}, \mathbf{z}, t\right)$
**end for**
**Output:** $\mathbf{x}_0^{(k+1)}$

---

A.3    INTEGRATION FLOW FOR PFGM++

The backward ODE of PFGM++ is characterized as:

$$\frac{d\mathbf{x}}{dr} = \frac{\mathbf{E}(\tilde{\mathbf{x}})_{\mathbf{x}}}{E(\tilde{\mathbf{x}})_r} \tag{8}$$

Since (Xu et al., 2023) showed that $d\mathbf{x}/dr = d\mathbf{x}/d\sigma$,where $\sigma$ changes with time.

We modify the equation 8 as:

$$\frac{d\mathbf{x}}{dt} = \frac{d\mathbf{x}}{dr}\frac{d\sigma_t}{dt} = \frac{\mathbf{E}(\tilde{\mathbf{x}})_{\mathbf{x}}}{E(\tilde{\mathbf{x}})_r}\frac{d\sigma_t}{dt}$$

We do the reversed time integration on both sides with respect to $t$ over the interval $[0, t]$, leading to:

$$\int_t^0 \frac{d\mathbf{x}}{dt}dt = \int_t^0 \frac{\mathbf{E}(\tilde{\mathbf{x}})_{\mathbf{x}}}{E(\tilde{\mathbf{x}})_r}\frac{d\sigma_t}{dt}dt,$$

which is equivalent to:

$$\mathbf{x}_0 - \mathbf{x}_t = \mathbf{V}\left(\mathbf{x}_0, 0\right) - \mathbf{V}\left(\mathbf{x}_t, t\right) = -F(\mathbf{x}_0, \mathbf{x}_t, t)$$

Rearranging the equation, we express the initial data point in terms of the integration flow:

$$\mathbf{x}_0 = \mathbf{x}_t - F(\mathbf{x}_0, \mathbf{x}_t, t) = \mathbf{f}\left(\mathbf{x}_0, \mathbf{x}_t, t\right).$$

For stable training purpose, we rewrite $\mathbf{f}\left(\mathbf{x}_0, \mathbf{x}_t, t\right)$ as following:

$$\mathbf{f}\left(\mathbf{x}_0, \mathbf{x}_t, t\right) = \mathbf{x}_t - F\left(\mathbf{x}_0, \mathbf{x}_t, t\right)$$
$$= a_t \mathbf{x}_t + (1 - a_t)\left[\mathbf{x}_t - \frac{1}{1 - a_t}F\left(\mathbf{x}_0, \mathbf{x}_t, t\right)\right]$$

Since $\mathbf{x}_t = \mathbf{x}_0 + R_t \mathbf{v}_t$, inspired by the settings of $a_t, b_t$ in VE case of diffusion model, we set $a_t = \sigma_{\min}/R_t, b_t = 1 - a_t = 1 - \sigma_{\min}/R_t$.

---

**Algorithm 5** Integration Flow Training Algorithm for PFGM++

---

**Input:** $p_{\text{data}}, T$, model parameter $\boldsymbol{\theta}$, initialize $\mathbf{x}_0^{(0)} \sim \mathcal{N}(\mathbf{0}, \mathbf{I})$, epoch $n \leftarrow 0$
**repeat**
    Sample $\mathbf{x}_0 \sim p_{\text{data}}, t \sim \mathcal{U}\left[1, T\right]$ and $R_t \mathbf{v}_t$, where $r_t = \sigma_t \sqrt{D}, R_t \sim p_{r_t}(R), \mathbf{v}_t = \mathbf{u}_t / \left\|\mathbf{u}_t\right\|_2, \mathbf{u}_t \sim \mathcal{N}(\mathbf{0}, \mathbf{I})$
    $\mathbf{x}_t = \mathbf{x}_0 + R_t \mathbf{v}_t$
    $\mathbf{x}_0^{(n+1)} \leftarrow \mathbf{f}_{\boldsymbol{\theta}}\left(\mathbf{x}_0^{(n)}, \mathbf{x}_t, t\right)$
    $\mathcal{L}_{\text{IF}}^{(n+1)}(\boldsymbol{\theta}) \leftarrow d\left(\boldsymbol{f}_{\boldsymbol{\theta}}\left(\mathbf{x}_0^{(n)}, \mathbf{x}_t, t\right), \mathbf{x}_0\right)$
    $\boldsymbol{\theta} \leftarrow \boldsymbol{\theta} - \eta \nabla_{\boldsymbol{\theta}} \mathcal{L}(\boldsymbol{\theta})$
    $n \leftarrow n + 1$
**until** convergence

---

**Algorithm 6** Integration Flow Sampling Algorithm for PFGM++

---

**Input:** $T$, trained model parameter $\boldsymbol{\theta}$, sampling step $k, r_T = \sigma_T \sqrt{D}, R_T \sim p_{r_T}(R), \mathbf{v} = \frac{\mathbf{u}}{\|\mathbf{u}\|_2}$, with $\mathbf{u} \sim \mathcal{N}(\mathbf{0}, \mathbf{I})$, initialize $\mathbf{x}_0^{(0)} \sim \mathcal{N}(\mathbf{0}, \mathbf{I})$
$\mathbf{x}_T = R_T \mathbf{v}$
**for** $k = 0$ **to** $k - 1$ **do**
    $\mathbf{x}_0^{(k+1)} \leftarrow \mathbf{f}_{\boldsymbol{\theta}}\left(\mathbf{x}_0^{(k)}, \mathbf{x}_T, T\right)$
**end for**
**Output:** $\mathbf{x}_0^{(n+1)}$

---

# B  PROOF OF THEOREMS

## B.1  PROOF OF **THEOREM 1**:

*Proof.* Before proving the theorem, we prove the corollary first:

**Corollary**:

$$\mathbb{E}\left[(A - \mathbb{E}[A \mid B, C])^2\right] \leq \mathbb{E}\left[(A - \mathbb{E}[A \mid B])^2\right],$$

The variance of a random variable $A$ can be decomposed as follows:

$$\text{Var}(A) = \mathbb{E}[\text{Var}(A \mid B)] + \text{Var}(\mathbb{E}[A \mid B]).$$

Now, introduce the extra information $C$. The variance of $A$, conditioned on both $B$ and $C$, is:

$$\text{Var}(A|B,C) = \mathbb{E}[\text{Var}(A|B,C)|B] + \text{Var}(\mathbb{E}[A|B,C]|B)$$

Since conditioning on more information reduces uncertainty, we know that:

$$\text{Var}(A|B,C) \leq \text{Var}(A|B).$$

We also have:

$$\mathbb{E}\left[(A - \mathbb{E}[A|B,C])^2\right] = \mathbb{E}[\text{Var}(A|B,C)]$$

and

$$\mathbb{E}\left[(A - \mathbb{E}[A|B])^2\right] = \mathbb{E}[\text{Var}(A|B)]$$

Since $\text{Var}(A|B,C) \leq \text{Var}(A|B)$, it follows that:

$$\mathbb{E}\left[(A - \mathbb{E}[A|B,C])^2\right] \leq \mathbb{E}\left[(A - \mathbb{E}[A|B])^2\right]$$

let $\mathbf{x}_0 = A, \mathbf{x}_t = B, \mathbf{x}_0^{(n)} = C$, plug into the corollary, we complete the proof $\qquad \square$

The following is the proof of any convex metric $d(\cdot, \cdot)$.

*Proof.* $\hat{\mathbf{x}}_0 = \mathbb{E}\left[\mathbf{x}_0 \mid \mathbf{x}_t\right]$ : The conditional expectation of $\mathbf{x}_0$ given $\mathbf{x}_t$

$\tilde{\mathbf{x}}_0 = \mathbb{E}\left[\mathbf{x}_0 \mid \mathbf{x}_t, \mathbf{x}_0^{(n)}\right]$ : The conditional expectation of $\mathbf{x}_0$ given both $\mathbf{x}_t$ and additional information $\mathbf{x}_0^{(n)}$.

$d\left(\mathbf{x}_0, a\right)$ is convex in $a$. The $\sigma$-algebra (information set) generated by $\left(\mathbf{x}_t, \mathbf{x}_0^{(n)}\right)$ is denoted by $\mathcal{F}$, and that generated by $\mathbf{x}_t$ is denoted by $\mathcal{G}$. Thus, $\mathcal{F} \supseteq \mathcal{G}$.

For a convex loss function $d$, the conditional expectation $\mathbb{E}\left[\mathbf{x}_0 \mid \mathcal{I}\right]$ minimizes the expected loss $\mathbb{E}\left[d\left(\mathbf{x}_0, a\right) \mid \mathcal{I}\right]$ over all $a$ measurable with respect to the information set $\mathcal{I}$. Therefore:

$$\tilde{\mathbf{x}}_0 = \mathbb{E}\left[\mathbf{x}_0 \mid \mathcal{F}\right] \quad \text{minimizes} \quad \mathbb{E}\left[d\left(\mathbf{x}_0, a\right) \mid \mathcal{F}\right].$$
$$\hat{\mathbf{x}}_0 = \mathbb{E}\left[\mathbf{x}_0 \mid \mathcal{G}\right] \quad \text{minimizes} \quad \mathbb{E}\left[d\left(\mathbf{x}_0, a\right) \mid \mathcal{G}\right].$$

Since $\mathcal{F} \supseteq \mathcal{G}$, conditioning on $\mathcal{F}$ provides at least as much information as conditioning on $\mathcal{G}$.

Next, we use Jensen's Inequality for Conditional Expectations.

For a convex function $d$, and any estimator $a$ measurable with respect to $\mathcal{G}$,

$$\mathbb{E}\left[d\left(\mathbf{x}_0, a\right) \mid \mathcal{F}\right] \geq d\left(\mathbb{E}\left[\mathbf{x}_0 \mid \mathcal{F}\right], a\right) = d\left(\tilde{\mathbf{x}}_0, a\right)$$

Since $\tilde{\mathbf{x}}_0$ minimizes $\mathbb{E}\left[d\left(\mathbf{x}_0, a\right) \mid \mathcal{F}\right]$,

$$\mathbb{E}\left[d\left(\mathbf{x}_0, \tilde{\mathbf{x}}_0\right) \mid \mathcal{F}\right] \leq \mathbb{E}\left[d\left(\mathbf{x}_0, a\right) \mid \mathcal{F}\right] \quad \forall a \text{ measurable with respect to } \mathcal{F}$$

Specifically for $a = \hat{\mathbf{x}}_0$ :

$$\mathbb{E}\left[d\left(\mathbf{x}_0, \tilde{\mathbf{x}}_0\right) \mid \mathcal{F}\right] \leq \mathbb{E}\left[d\left(\mathbf{x}_0, \hat{\mathbf{x}}_0\right) \mid \mathcal{F}\right]$$

Taking the expectation over both sides with respect to the entire probability space,

$$\mathbb{E}\left[d\left(\mathbf{x}_0, \tilde{\mathbf{x}}_0\right)\right] \leq \mathbb{E}\left[d\left(\mathbf{x}_0, \hat{\mathbf{x}}_0\right)\right]$$

we complete the proof. $\qquad \square$

## B.2 Proof of **Theorem 2**:

*Proof.* The initial value problem (IVP) of the reversed time ODE can be expressed as:

$$\begin{cases} \dfrac{d\mathbf{x}_s}{ds} = \mathbf{v}\left(\mathbf{x}_s, s\right) & s \in [0, t] \\ \mathbf{x}_t = \hat{\mathbf{x}}_t \end{cases} \tag{9}$$

if putting $\tilde{\mathbf{x}}_s := \mathbf{x}_{t-s}$, we get

$$\begin{cases} \dfrac{d\tilde{\mathbf{x}}_s}{ds} = -\mathbf{v}\left(\tilde{\mathbf{x}}_s, s\right) & s \in [0, t] \\ \tilde{\mathbf{x}}_0 = \hat{\mathbf{x}}_t \end{cases} \tag{10}$$

The IVP 9 and 10 are equivalent and can be used interchangeably.

From the Lipschitz condition on $\mathbf{v}$, we have:

$$\|\mathbf{v}(\tilde{\mathbf{x}}_s, s) - \mathbf{v}(\tilde{\mathbf{y}}_s, s)\|_2 \le L \|\tilde{\mathbf{x}}_s - \tilde{\mathbf{y}}_s\|_2 .$$

Use the integral form:

$$\|\mathbf{f}\left(\mathbf{x}_0, \mathbf{x}_t, t\right) - \mathbf{f}\left(\mathbf{y}_0, \mathbf{y}_t, t\right)\|_2 \le \|\tilde{\mathbf{x}}_0 - \tilde{\mathbf{y}}_0\|_2 + \int_0^t L \|\tilde{\mathbf{x}}_s - \tilde{\mathbf{y}}_s\|_2 \, ds$$

By using Gröwnwall inequality, we have:

$$\|\mathbf{f}\left(\mathbf{x}_0, \mathbf{x}_t, t\right) - \mathbf{f}\left(\mathbf{y}_0, \mathbf{y}_t, t\right)\|_2 \le e^{Lt} \|\tilde{\mathbf{x}}_0 - \tilde{\mathbf{y}}_0\|_2 = e^{Lt} \|\hat{\mathbf{x}}_t - \hat{\mathbf{y}}_t\|_2$$

Next, consider the inverse time ODE, we have:

$$\|\mathbf{x}_t - \mathbf{y}_t\|_2 \le \|\mathbf{f}(\mathbf{x}_0, \mathbf{x}_t, t) - \mathbf{f}(\mathbf{y}_0, \mathbf{y}_t, t)\|_2 + \int_0^t L \|\mathbf{x}_s - \mathbf{y}_s\|_2 \, ds$$

Again, by using Gröwnwall inequality,

$$\|\mathbf{x}_t - \mathbf{y}_t\|_2 \le e^{Lt} \|\mathbf{f}(\mathbf{x}_0, \mathbf{x}_t, t) - \mathbf{f}(\mathbf{y}_0, \mathbf{y}_t, t)\|_2$$

Therefore,

$$\|\mathbf{f}\left(\mathbf{x}_0, \mathbf{x}_t, t\right) - \mathbf{f}\left(\mathbf{y}_0, \mathbf{y}_t, t\right)\|_2 \ge e^{-Lt} \|\mathbf{x}_t - \mathbf{y}_t\|_2$$

and we complete the proof of:

$$e^{-Lt} \|\mathbf{x}_t - \mathbf{y}_t\|_2 \le \|\mathbf{f}(\mathbf{x}_0, \mathbf{x}_t, t) - \mathbf{f}(\mathbf{y}_0, \mathbf{y}_t, t)\|_2 \le e^{Lt} \|\mathbf{x}_t - \mathbf{y}_t\|_2 . \tag{11}$$

Since the neural network is sufficiently trained, $\mathbf{f}_{\boldsymbol{\theta}^*}\left(\mathbf{x}_0^{(n)}, \mathbf{x}_t, t\right) \equiv \mathbf{f}\left(\mathbf{x}_0, \mathbf{x}_t, t\right)$, replace $\mathbf{f}\left(\mathbf{x}_0, \mathbf{x}_t, t\right), \mathbf{f}\left(\mathbf{y}_0, \mathbf{y}_t, t\right)$ with $\mathbf{f}_{\boldsymbol{\theta}^*}\left(\mathbf{x}_0^{(n)}, \mathbf{x}_t, t\right), \mathbf{f}_{\boldsymbol{\theta}^*}\left(\mathbf{y}_0^{(n)}, \mathbf{y}_t, t\right)$ respectively in equation 11, we obtain:

$$e^{-Lt} \|\mathbf{x}_t - \mathbf{y}_t\|_2 \le \left\|\mathbf{f}_{\boldsymbol{\theta}^*}\left(\mathbf{x}_0^{(n)}, \mathbf{x}_t, t\right) - \mathbf{f}_{\boldsymbol{\theta}^*}\left(\mathbf{y}_0^{(n)}, \mathbf{y}_t, t\right)\right\|_2 \le e^{Lt} \|\mathbf{x}_t - \mathbf{y}_t\|_2$$

$\square$

## C   ADDITIONAL SAMPLES

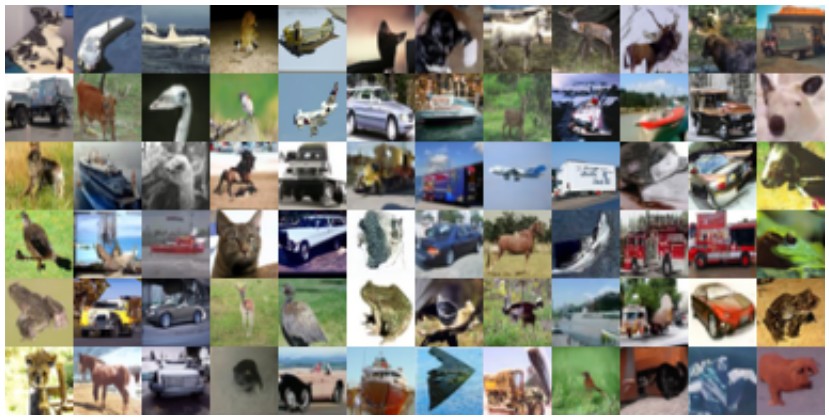

Figure 5: One-step samples from the Integration Flow-VE model (FID=2.87)

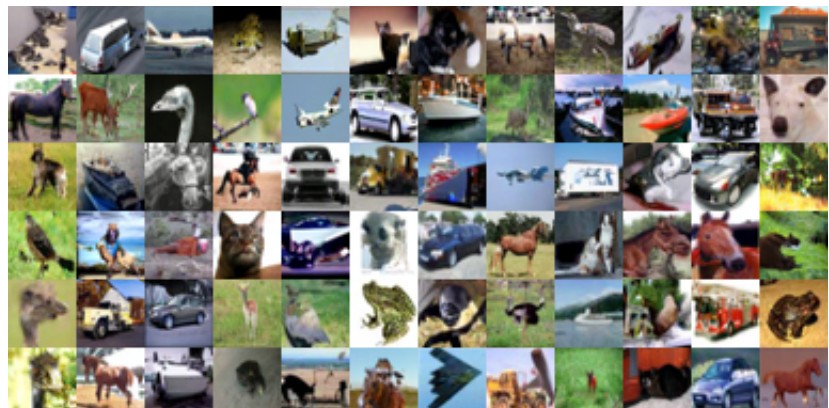

Figure 6: Two-step samples from the Integration Flow-VE model (FID=2.64)

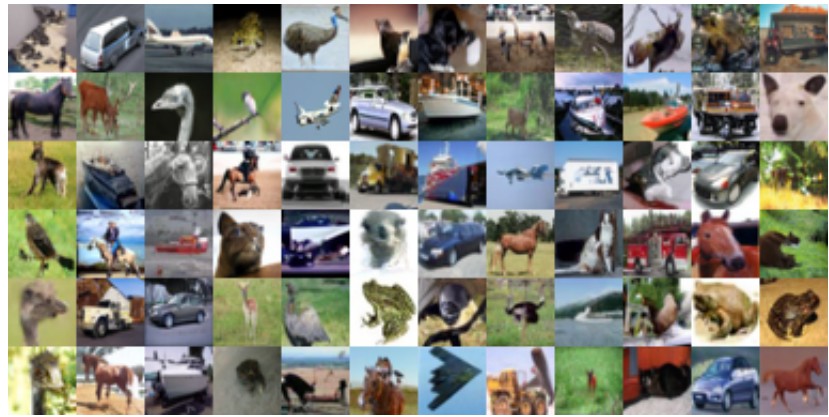

Figure 7: 1000-step samples from the Integration Flow-VE model (FID=1.89)

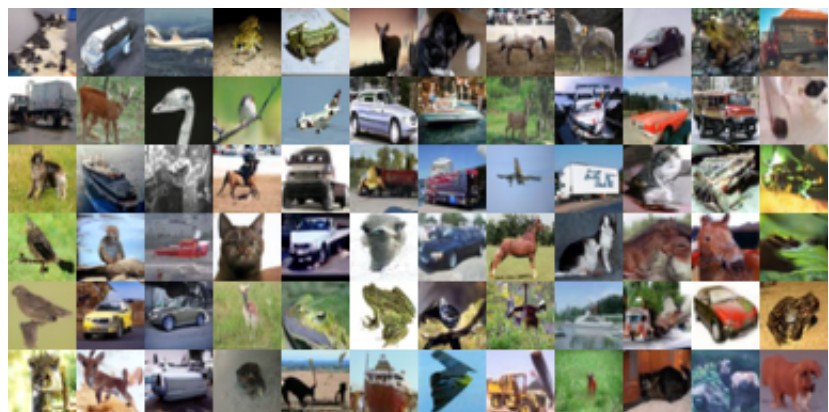

Figure 8: One-step samples from the Integration Flow-VE model (FID=2.63).

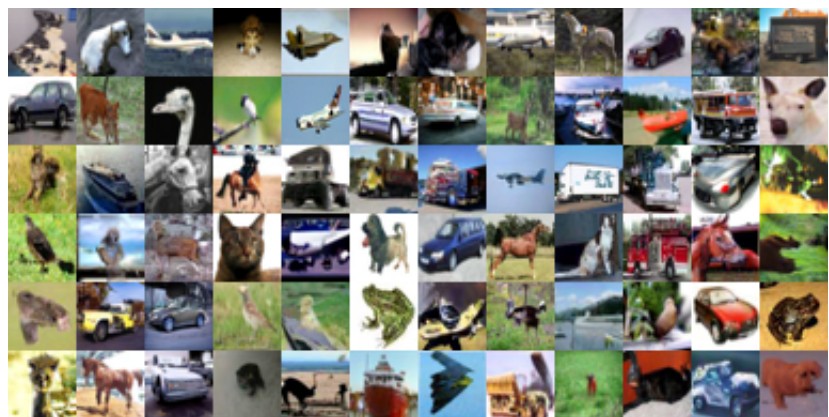

Figure 9: Two-step samples from the Integration Flow-VE model (FID=2.35).

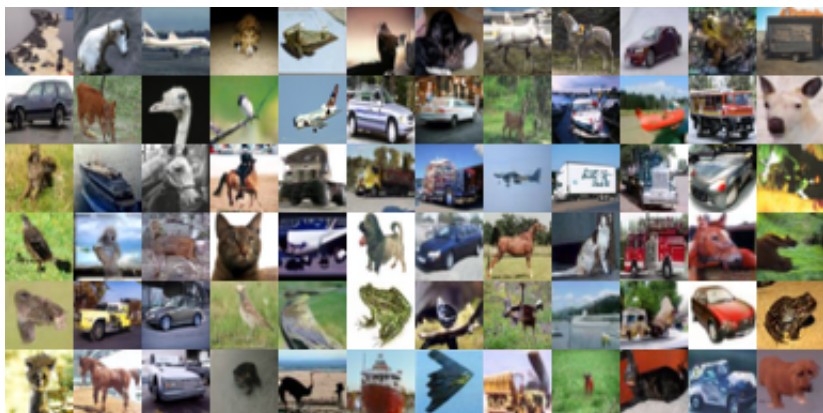

Figure 10: 1000-step samples from the Integration Flow-VE model (FID=1.71).

