# OpenReview forum: "Integration Flow Models"
_ICLR.cc/2025/Conference — ICLR 2025 Conference Withdrawn Submission_

### Official Review · Reviewer_sT29 · 2024-10-27

**Soundness:** 2
**Presentation:** 2
**Contribution:** 2
**Rating:** 5
**Confidence:** 3

**Summary:**

The authors introduced Integration Flow which is an ODE-based generative model that learns the trajectory results directly without solving ODE functions, thereby addressing discretization errors and training instability in traditional methods. By explicitly incorporating the target state as an anchor for guiding reverse-time dynamics, it enhances both stability and accuracy in sample generation. Empirical results show that Integration Flow achieves state-of-the-art performance on CIFAR10, with low FID scores in one-step generation and further improvements when extended to multiple steps.

**Strengths:**

1. The paper is easy to understand.
2. The idea is interesting

**Weaknesses:**

There is no significant drawback of this paper in terms of presentation. However:

I find the claim of being the first *unified* ODE generative model questionable. If the authors refer to a unified training and sampling scheme, then the Stochastic Interpolant paper [1] has already addressed this effectively. If they are referring to techniques for directly estimating the ODE solution, then BOOT [2] has also shown promising results.

Perhaps I have misunderstood some parts of the algorithm section. I will raise a few questions regarding this in the question section.

It would be beneficial if the authors could draw some connections with the consistency model. Intuitively, they are closely related.

[1] Song, Yang, et al. "Consistency models." arXiv preprint arXiv:2303.01469 (2023).

[2] Gu, Jiatao, et al. "Boot: Data-free distillation of denoising diffusion models with bootstrapping." ICML 2023 Workshop on Structured Probabilistic Inference {\&} Generative Modeling. 2023.

**Questions:**

1. Is there any connection with Consistency Model?

2.  In algorithm 1, The estimated data $x_0^{n+1}$ is never used during the training at $n$-th loop?

3. For example, if I sample a cat image $x_0$ at loop $n$, and then the estimated $x_0^{n}$ should be similar to a cat. For the $n+1$-th training loop, if I sample a $x_0$ from plane distribution, then I am still expecting $f_{\theta}(x_0^{n},\cdot,\cdot)$ will give me an estimation of a plane image with a cat image ($x_0^{n}$)? Does it make sense? Am I misunderstanding anything?

[1] Song, Yang, et al. "Consistency models." arXiv preprint arXiv:2303.01469 (2023).]

---

### Official Review · Reviewer_yuKx · 2024-10-30

**Soundness:** 2
**Presentation:** 2
**Contribution:** 3
**Rating:** 5
**Confidence:** 4

**Summary:**

The paper proposes integration flow models, an approach to bypass ODE solvers in flow-based generative models. Integration Flow claims that explicitly incorporating the target state as the anchor state in guiding the reverse-time dynamics provides stability and accuracy.

**Strengths:**

- The paper gives a unified framework for a variety of probability flow-based generative models.
- The intuition of explicitly having the target state as the anchor is theoretically well-justified.
- The method achieves competitive FID scores with fewer generation steps on CIFAR-10.

**Weaknesses:**

- The training algorithm is not well presented and there is no discussion about its correctness.
- The paper shows experimental results for the CIFAR-10 dataset only. While CIFAR-10 is a standard benchmark, the lack of experiments on larger or more diverse datasets (such as ImageNet or higher-resolution data) raises questions about the generalizability of the method to other domains.
- The authors acknowledge that the performance on VE diffusion models is slightly below the state-of-the-art due to sub-optimal hyper-parameters.

**Questions:**

- For the training phase $x_0^{(0)} \sim N(0, I)$. For each subsequent $n$, a fresh $(x_0, z)$ is sampled from $p_{\text{data, } z}$ (and hence, a fresh $x_t$) but the same $x_0^{(n)}$ is updated as $x_0^{(n+1)} \leftarrow f_\theta(x_0^{(n)}, x_t, t)$ until convergence.

    -- What does $\lim_{n \to \infty}x_0^{(n)}$ converge to?

    -- How do we know if the algorithm even converges?

    -- What is the model $f_\theta$ really anchoring towards? This question is more focused on the intuition behind this approach.

    -- Alternatively, why shouldn't $x_0^{(0)}$ be sampled independently for all new $(x_0, z)$ pairs, and the iterative update be done until convergence?
- As a follow up of the previous question, for one step sampling, consider the example of rectified flow: the algorithm suggests to have $x_0 = f_\theta(x_0^{(0)}, z, t=1)$, where $x_0^{(0)}, z \sim N(0, I)$ (assuming the base distribution to be a Gaussian). What does a random Gaussian draw anchor the model toward, if the training is done as suggested in the paper?
- Is it possible to show some empirical evidence of convergence of this algorithm on different datasets? Maybe just checking on simple simulated datasets if $\lim_{n \to \infty}x_0^{(n)}$ is some meaningful statistic?
- Lastly, is it possible to do experiments on more higher-resolution datasets to see the generalizability of the method?

---

### Official Review · Reviewer_cngm · 2024-11-01

**Soundness:** 2
**Presentation:** 2
**Contribution:** 2
**Rating:** 1
**Confidence:** 4

**Summary:**

This paper proposes the Integration Flow, a framework that allows both one-step sampling and multi-step sampling for ODE-based methods like diffusion model, rectified flow, and Poisson flow generative models.

**Strengths:**

1. The experiment results are competitive compared with other diffusion models, flow-based methods, and distillation methods.

**Weaknesses:**

1. There are concerns that this paper may have **plagiarized content** from the paper [1]. First, the motivation and formulation in this paper (Sec 3.1, 3.2, 3.3) are quite similar to Section 3 in [1]. The training algorithm and sampling algorithm are almost the same (Algorithm 1, 3 in this paper vs Algorithm 1 in [1]  $\quad$ and $\quad$   Algorithm 2, 4 in this paper vs Algorithm 2 in [1]). However, I do not find any citation or discussion of [1] in this paper.

2. The proof of the paper contains technical flaws. For example, in line 840, the author claims that $E[x_0 | I]$ is the minimizer of $E[d(x_0, a)| I]$ for a if $d$ is a convex function. However, a basic fact is that l1 loss is convex and the minimizer of l1 loss is conditional **median** instead of conditional **mean**.




[1] Directly Denoising Diffusion Models. arxiv 2405.13540

**Questions:**

I do not have other questions.

**Details Of Ethics Concerns:**

There are concerns that this paper may have **plagiarized content** from the paper [1]. First, the motivation and formulation in this paper (Sec 3.1, 3.2, 3.3) are quite similar to Section 3 in [1]. The training algorithm and sampling algorithm are almost the same (Algorithm 1, 3 in this paper vs Algorithm 1 in [1]  $\quad$ and $\quad$   Algorithm 2, 4 in this paper vs Algorithm 2 in [1]). However, I do not find any citation or discussion of [1] in this paper.


[1] Directly Denoising Diffusion Models. arxiv 2405.13540

---

> ### Author Response · Authors · 2024-11-13
> **There is no plagiarized content!**
>
> Dear reviewer cngm,
>
> We **strongly disagree** with your charge about the plagiarized content. This is NOT in line with the facts. This paper extends our previous DDDM paper to the general ordinary differential equation (ODE)—based generative models. In DDDM, we only worked on the diffusion model of variance preservation (VP). In this paper, we extend it to more general ODE-based generative models, including diffusion models (VE), Rectified Flows, and PFGM++. Although the algorithm formulation is similar, we have extended it to a more general format and the writing is different. This can not be called plagiarised. We strongly request you remove the plagiarism charge.
>
> Sincerely Yours
> Submission3267 authors

---

### Official Review · Reviewer_cHtf · 2024-11-04

**Soundness:** 3
**Presentation:** 2
**Contribution:** 2
**Rating:** 5
**Confidence:** 3

**Summary:**

Sampling from certain kinds of performant generative models (e.g., diffusion models and Poisson flow generative models) involves integrating an ordinary differential equation (ODE), which can be computationally costly, and produce downstream issues via compounding discretization errors if simple or coarse integration schemes are used. The authors propose a neural-network-based approach to replace the ODE integration step, and hence save computation at the time of sample generation. They apply their approach to speed up sample generation in three kinds of generative models: diffusion models, Poisson flow generative models, and rectified flow models. Their approach yields near-SOTA performance in a few cases, especially given the constraint of using a small number of function evaluations.

**Strengths:**

The authors propose a reasonable scheme for replacing ODE integration steps that seems to perform relatively well in practice, and show that it yields good performance for a few kinds of generative models. The idea is relatively easy to follow.

**Weaknesses:**

My major concern is about the novelty of the paper. Replacing ODE integration with something else, including some kind of neural network, is not a new idea, and from the authors' exposition their contribution relative to the contribution of previous work is unclear. One related idea is to use an ANN to learn 'coarser' integration steps (see, e.g., Huang et al., 2023, https://www.nature.com/articles/s41598-023-42194-y, although there are other papers of this kind too). Another idea, which the authors mentioned in a performance comparison but not in a conceptual comparison, is that of consistency models, which replace the ODE step of diffusion models with a more explicit learned map from the latent space to the learned distribution space. Some discussion of how the authors' proposal relates to these and other proposals, and what the specific novelty is, would be helpful. If the key contribution of the authors is to implement a familiar idea in a more efficient or performant way, they should state this explicitly.

Another concern regards the reliability of the proposed approach. The authors use two theorems (Sec. 3.4) to address this, but the theorems don't really concern that much the details of the authors' proposed approach. For example, Theorem 2 is just a statement about (true) ODE trajectories being unique, which is a well-known classic result. Numerical integration methods and their drawbacks are well-understood, and when one replaces them with neural networks one loses certain nice theoretical guarantees. What should one be careful of if one uses Integration flow? If a certain ODE is somewhat out-of-distribution, does it not get integrated properly? How badly can things fail? An analysis of this kind is crucial if the proposed method is to be useful. Said differently, what can be said about the types of errors this method tends to produce, analogous to the discretization errors of numerical integration schemes?

More minor concerns involve typos or clarity. Some typos: line 64, needs period; line 107, needs space; line 305, "conculde". Clarity: line 135, what is $v$? Fig. 4: what is $N$? I think it's the number of iterations used (c.f. line 252) but this should be stated near the figure somewhere.

It is mentioned (line 468) that Integration Flow 'unifies' different types of generative models, but this is kind of silly. Integration Flow is not a theoretical framework but a tool for replacing ODE integration. The types of generative models mentioned are only unified in the sense that they all involve ODEs, which is not particularly unified, and true regardless of Integration Flow's existence.

A nitpick: the review of existing generative model ODEs could be reorganized. I would put Sec. 3 before Sec. 2, so that it's clearer what the author's contribution is versus what are used as examples.

**Questions:**

1. How does the authors' proposal compare to other ideas for replacing numerical integration, including the consistency model idea and analogous 'ODE coarse-graining' ideas? Are these things pretty similar, and mostly the details are what's different? Are they fundamentally different?
2. Can the authors justify the reliability of their approach? What kind of errors might users encounter, and how do these compare to well-understood discretization errors? When might a user be in an 'out-of-distribution' setting?

---

### Note · Authors · 2024-11-14

I have read and agree with the venue's withdrawal policy on behalf of myself and my co-authors.